# Are We Getting It Right? A Scoping Review of Outcomes Reported in Cell Therapy Clinical Studies for Cerebral Palsy

**DOI:** 10.3390/jcm11247319

**Published:** 2022-12-09

**Authors:** Megan Finch-Edmondson, Madison C. B. Paton, Ingrid Honan, Petra Karlsson, Candice Stephenson, Darryl Chiu, Sarah Reedman, Alexandra R. Griffin, Catherine Morgan, Iona Novak

**Affiliations:** 1Cerebral Palsy Alliance Research Institute, Speciality of Child and Adolescent Health, Sydney Medical School, Faculty of Medicine and Health, The University of Sydney, Sydney, NSW 2050, Australia; 2Faculty of Medicine and Health, The University of Sydney, Sydney, NSW 2050, Australia

**Keywords:** cerebral palsy, cell therapies, stem cells, comorbidities, outcome measures, clinical studies

## Abstract

Cell therapies are an emergent treatment for cerebral palsy (CP) with promising evidence demonstrating efficacy for improving gross motor function. However, families value improvements in a range of domains following intervention and the non-motor symptoms, comorbidities and complications of CP can potentially be targeted by cell therapies. We conducted a scoping review to describe all outcomes that have been reported in cell therapy studies for CP to date, and to examine what instruments were used to capture these. Through a systematic search we identified 54 studies comprising 2066 participants that were treated with a range of cell therapy interventions. We categorized the reported 53 unique outcome instruments and additional descriptive measures into 10 categories and 12 sub-categories. Movement and Posture was the most frequently reported outcome category, followed by Safety, however Quality of Life, and various prevalent comorbidities and complications of CP were infrequently reported. Notably, many outcome instruments used do not have evaluative properties and thus are not suitable for measuring change following intervention. We provide a number of recommendations to ensure that future trials generate high-quality outcome data that is aligned with the priorities of the CP community.

## 1. Introduction

Stem cell and cell therapies have been in clinical research for the treatment of cerebral palsy (CP) for more than 15 years [1]. There are a variety of cell types being investigated including umbilical cord blood, mesenchymal stem/stromal cells, and neural stem- or stem-like cells [2]. The principal target of cell therapies for the treatment of CP is remediation of the underlying brain injury thereby improving neuronal signaling, which could be achieved by either direct or indirect actions. Cell therapies are proposed to work via a variety of mechanisms for the treatment of CP. Depending on the cell type, therapeutic benefits may include reduction of inflammation, promotion of cell survival, stimulation of proliferation and migration of endogenous neural stem cells, replacement and/or regeneration of damaged brain cells, and promotion of angiogenesis [2].

Systematic reviews of randomized controlled trials have shown that improvement in gross motor skills/function, typically measured using the Gross Motor Function Measure (GMFM) [3], is the most common primary outcome assessed [4,5]. Promisingly, these studies have demonstrated that various cell therapies can produce a small but significant improvement in gross motor function [4,5], although these findings are limited by heterogeneity in various aspects (e.g., participants, interventions, outcomes). Furthermore, whilst the number of clinical studies and total number of participants with CP treated with cell therapies continues to climb (now >2427 participants across >77 published and unpublished studies) [1], there remains a high volume of lower-quality evidence employing poor study design and/or unvalidated outcome assessment tools, and thus more research is warranted.

Although CP is characterized by impairment of movement and/or posture, it is a highly heterogeneous condition, and individuals with CP often experience a range of comorbidities and/or co-occurring complications that can be just as disabling as the motor symptoms [6]. These include pain, intellectual impairment, epilepsy, behavior disorders, and vision and hearing impairments [6]. As such, there is an increasing focus within the CP field to understand these elements, and find ways to target them, with the overarching goal of improving the quality of life for people living with CP.

Individuals with CP and their families value a wide range of potential benefits following certain types of stem cell treatments [7] and other interventions [8,9]. These benefits often focus on activity and participation rather than necessarily remediating physical impairment. It is therefore important that clinical studies of cell therapies measure outcomes that are both scientifically valid and valued by individuals with CP and their families. In addition, outcomes should be measured using well-validated tools so that evidence generated from these studies can increase our confidence in study findings. To aid in this, a panel of international experts have compiled recommended CP-specific common data elements for use in clinical research studies [10]. However, these instruments may not always be consistently applied. As such, the purpose of this scoping review is to describe all outcomes that have been reported in cell therapy studies for CP to date, and to examine what instruments have been used to capture these outcomes.

## 2. Materials and Methods

A protocol for this review was registered on Open Science Framework (OSF) (identifier DOI 10.17605/OSF.IO/T9C8J [https://osf.io/t9c8j/?view_only=9b82c37725834a1da1a50bb199cf5091 (accessed on 14 November 2022)], registration date 8 July 2022). This scoping review was conducted according to the Preferred Reporting Items for Systematic Reviews and Meta-Analyses extension for Scoping Reviews (PRISMA-ScR) guidelines [11] (Appendix A).

### 2.1. Inclusion and Exclusion Criteria

We included any type of study (both controlled and non-controlled studies, including case series/reports) in which participants with CP were treated with a cell therapy intervention specifically for the treatment of CP. If studies reported participants with various diagnoses, >50% must have had CP. There was no restriction on participant age. The full text of the study must also have been published in English (due to no translation services available), in a peer-reviewed journal. Studies were excluded from this review if they reported an organ graft or transplant, or were a secondary analysis of a study that was already included in this review.

### 2.2. Data Sources and Search Strategy

We searched the Cochrane Central Register of Controlled Trials (CENTRAL) (The Cochrane Library, April 2022), PubMed (MEDLINE) (1946 to 6 May 2022) and EMBASE (1947 to 6 May 2022) via OVID using the search strategy described in Appendix A. The search was conducted on 10th May 2022. De-duplicated results from OVID were exported into Covidence Systematic Review Software (http://www.covidence.org (accessed on 14 November 2022)). Database searching was also supplemented by hand searching, i.e., cross checking systematic review reference lists for potentially eligible articles, and new paper alerts were monitored for potentially eligible papers published after the formal search was conducted.

### 2.3. Study Selection and Data Extraction

Titles and/or abstracts of studies retrieved using the search strategy were screened independently by two reviewers (split between M.F.-E., M.C.B.P., C.F.). Full texts of studies were then retrieved and independently assessed for eligibility by two reviewers (split between M.F.-E., M.C.B.P., C.F.), with any disagreements resolved by the third screener.

A data extraction form was developed specifically for this review by the research team. Data extraction was performed independently by at least two members of the research team (M.F.-E., M.C.B.P., C.F.), with any discrepancies identified and resolved through discussion with the third extractor. Extracted data included details of study design, participants, intervention/s, comparator (if relevant), and outcome instrument/s.

### 2.4. Assigning Level of Evidence for Included Studies

The level of evidence for each included study was assigned according to Oxford Centre for Evidence-Based Medicine: Levels of Evidence [12].

### 2.5. Categorization of Cell Interventions

Cell interventions were sorted into six categories: (1) Umbilical cord blood; (2) Mesenchymal stem/stromal cells; (3) Bone marrow cells, hematopoietic stem cells and peripheral blood cells (including mononuclear cell fragment, enriched/expanded cells from bone marrow or umbilical cord blood, and peripheral blood mononuclear cells); (4) Neural stem cells/neural-like cells (including neural stem cells (NSCs), neural progenitor cells, olfactory ensheathing cells and mesenchymal stem/stromal cell-derived NSC-like cells); (5) Immune cells (M2-like macrophages); and (6) Fetal cells/embryonic stem cells.

### 2.6. Categorization of Instruments (into Outcome Domains)

For this review, members of the research team (M.F.-E., M.C.B.P., I.H., P.K., C.S., D.C.) determined outcome domain categories and sub-categories for sorting the reported outcome instruments. This process involved consideration of all the extracted outcome instruments followed by group discussion to reach consensus on which outcome categories/sub-categories to include. All outcome instruments were then assigned to these categories/sub-categories according to the outcome domain/s they were designed to assess, again via group discussion between multiple members of the research team (M.F.-E., M.C.B.P., P.K., I.H., C.S., D.C.) to reach agreement.

Outcome instruments that spanned more than one outcome domain, i.e., encompassing multiple reported sub-domains, were assigned to various categories/sub-categories according to these sub-domains. Any instrument for which the outcome being assessed could not be determined (or agreed), the tool was a multi-domain measure but was only reported as a total score, or the instrument did not fit with any other outcome sub/category, were designated as Other. Any reported descriptive/observational outcomes were subsequently categorized into the same outcome sub/categories through discussion and agreement.

### 2.7. Outcome Instrument Properties

Outcome instrument properties including format (i.e., who completed the assessment and the nature of it), primary purpose (i.e., predictive, discriminative, evaluative or classification) and population designed for, were determined from various information sources including test manuals/handbooks, systematic reviews, and websites, as necessary.

The categories used for instrument format followed that of the U.S. Food and Drug Administration (FDA) types of clinical outcome assessments, namely: Patient (or self)-reported, Clinician (or therapist)-reported, Observer (e.g., parent/carer/teacher)-reported, and Performance-based measures [13]. In this review we have used the term ‘Parent/other’ to denote the Observer group. Additionally, the report type was specified as Questionnaire, Interview (including semi-structured interview) or Observation.

### 2.8. Calculating Total Number of Participants per Outcome Sub/Category

For calculating the number of participants assessed for each outcome sub/category, n’s were collapsed or compounded as such: In studies that utilized multiple assessment tools within a single outcome sub-category (e.g., *Gross Motor* measured using GMFM, Gross Motor Function Classification System Expanded and Revised (GMFCS) [14], etc.), the number of participants assessed for *Gross Motor* was collapsed, meaning that n’s were not counted more than once for that sub-category. E.g., if 10 participants were evaluated using the GMFCS and GMFM, the n for *Gross Motor* would remain at 10. However, if the same 10 study participants were also assessed for *Fine Motor and Upper Limb* using the Fine Motor Function Measure (FMFM) [15], this would result in a total compounded n of 20 for the *Movement and Posture* category.

## 3. Results

### 3.1. Search Results

Following the literature search and de-duplication, 1145 records were identified. After title and abstract screening, 93 full-text reports were reviewed and 50 met eligibility [16,17,18,19,20,21,22,23,24,25,26,27,28,29,30,31,32,33,34,35,36,37,38,39,40,41,42,43,44,45,46,47,48,49,50,51,52,53,54,55,56,57,58,59,60,61,62,63,64,65]. A further four eligible reports were identified through hand searching [66,67,68,69]. In addition, during data extraction and preparation of the manuscript, two new studies were identified [70,71] that also met eligibility and were included. Moreover, a study that was initially included was retracted [30] and was therefore subsequently excluded from this review. Thus, finally, 55 reports were included. These 55 reports represented 54 studies since Amanat 2021 [17] and Zarrabi 2022 [64] are two reports of the same clinical trial (clinical trial registration identifier NCT03795974) and share the same control group. The PRISMA [72] flow chart of the search process is presented in Figure 1.

### 3.2. Study Characteristics

A summary of the included studies is presented in Table 1, including details of study design, participants, intervention/s and comparator/s, and outcome instrument/s.

### 3.3. Types of Studies

Of the 54 studies included, 17 (32%) were controlled studies: 14 were randomized controlled trials, with three of these being a cross-over design, and three were non-randomized controlled studies. A further 18 studies (33%) were single-arm and 18 (33%) were case series or case reports, including four studies [19,26,50,56] that were retrospective analyses including ‘therapeutic experiments’ and a ‘post-registration clinical investigation’. In addition, one study (2%) [27] was a non-randomized dose comparison trial. Accordingly, the majority of included studies (n = 37, 69%) were deemed to be Level 4 evidence with n = 3 being Level 3, and n = 14 were Level 2.

### 3.4. Types of Participants

Collectively, data from 2066 participants was reported, and all studies exclusively included participants with CP. Most studies enrolled/treated participants of various type and topography, and all GMFCS severity levels were represented. Whilst the majority of studies recruited/treated children and youth (up to 18 years) with CP, participant ages ranged from 6 months to 35 years (Table 1).

### 3.5. Types of Interventions

The majority of studies administered one cellular intervention, however four studies in five reports [17,36,64,66,71] investigated two different cell therapies head-to-head to give a total of 58 cell regimens administered.

For these 58 cell regimens, the classification into the various cell therapy types was: 31% bone marrow cells, hematopoietic stem cells and peripheral blood cells (n = 18); 29% mesenchymal stem/stromal cells (n = 17); 26% umbilical cord blood (n = 15); 7% neural stem cells/neural-like cells (n = 4); 3% immune cells (n = 2); and 3% fetal cells/embryonic stem cells (n = 2). The source of cells was autologous in 32 studies (55%) and allogeneic in 25 studies (43%). The donor origin of the cells could not be determined in one study (2%) [25].

Cell interventions were delivered by various routes. Intrathecal (n = 25, 43%) or intravenous (n = 20, 34%) delivery was the most common. A further three studies (5%) used a combination of the two. Exclusive direct transplantation into the brain (intracerebral, intra-cerebroventricular) was used infrequently (n = 3, 5%), and all were for studies that administered neural stem cells/neural-like cells. In addition, one study (2%) [61] used intrathecal +/− intra-parenchymal brain administration for mesenchymal stem/stromal cells. The remainder of the studies (n = 6, 10%) utilized various routes (or a combination of routes) including but not limited to intra-arterial or intramuscular delivery (Table 1).

### 3.6. Types of Outcome Measures

Instruments measuring treatment outcomes were grouped into ten categories: (1) Movement and Posture; (2) Cognition and General Development; (3) Communication and Language; (4) Behavior; (5) Activities of Daily Living; 6) Quality of Life; (7) Brain Structure and Function; (8) Biomarkers; (9) Safety and (10) Other. Four of these categories were further split into a total of 12 sub-categories (Table 1, Figure 2). The 12 sub-categories were Gross Motor, Fine Motor and Upper Limb, Spasticity, Muscle Strength and General Motor within Movement and Posture; Communication and Language within Communication and Language; Adaptive Behavior, Executive Function and Social-Emotional within Behavior; and Neuroimaging and Seizures/Electrical Brain Activity within Brain Structure and Function.

Unsurprisingly, *Movement and Posture* was the most frequently reported outcome category (n = 4195) (Figure 2). Indeed, all included studies reported on *Movement and Posture* except Feng 2015 [26], which exclusively reported safety. Within *Movement and Posture*, measures of *Gross Motor* were the most common, followed by *Fine Motor and Upper Limb* then *Spasticity*. *Safety* was the next most common category (n = 1705) and was specifically reported in all but 11 studies. Reported safety data included adverse event reporting, routine laboratory and clinical assessments (e.g., bloods/biochemistry, X-ray), and neuroimaging conducted exclusively for safety. *Brain Structure and Function* (n = 1083), *Behavior* (n = 1010) and *Activities of Daily Living* (n = 865) were all also commonly reported outcome categories (Figure 2). In contrast, a relatively small proportion of participants were assessed for *Biomarkers* and *Quality of Life*. Of the four studies that conducted biomarker analysis these comprised assessment of various cytokine and growth factor levels including interferon (IFN)-γ, interleukin (IL)-17, IL-4, brain-derived neurotrophic factor (BDNF), and vascular endothelial growth factor (VEGF) [22], BDNF [23], pentraxin 3 (PTX-3), IL-8, and IL-10 [33] and PTX-3, IL-8, tumor necrosis factor (TNF)-α, and IL-1β [41].

Examining the data by either study design or cell intervention type revealed a similar pattern, with *Movement and Posture* consistently the most frequently reported outcome category, followed by *Safety*, and a relatively similar distribution of participants across outcome sub/categories (Appendix A).

Across the included studies there were 53 unique instruments reported, although not all were true outcome measure, i.e., responsive to change (Table 1 and Table 2). This number does not include measures of *Safety* or *Biomarkers* since these are commonly reported in various ways and could not be synthesized, nor descriptive/observational outcomes. The categorization of all instruments into outcome sub-/categories is shown in Table 2. Notably, 12/53 of the captured instruments had multiple sub-domains that were reported and hence were included across several outcome categories/sub-categories in Table 2.

More than one instrument was used across the studies for the majority of outcome categories/sub-categories. For example, *Gross Motor* was assessed using 12 different tools, *Cognition and General Development* by 11, and *Activities of Daily Living* by eight different instruments (Table 2). In contrast, *Executive Function* was assessed using just a single instrument, the Behavior Rating Inventory of Executive Function (BRIEF) [74], in a single study.

The most commonly reported instrument was the GMFM (n = 1163), with this measure reported for 56% of all included participants in this review. The Pediatric Evaluation of Disability Inventory (PEDI)/PEDI-Computer Adaptive Test (PEDI-CAT) [75] (n = 573), GMFCS (n = 533) and magnetic resonance imaging (MRI) with or without diffusion tensor imaging (DTI) (n = 525) were also frequently used (Table 2).

Of note, study participants were often assessed using more than one instrument within an outcome category/sub-category (Table 1). This was particularly true for *Gross Motor*. For example, Rah 2017 [49] assessed participants using the GMFM, GMFCS, PEDI and Denver Development Screening Test (DDST) [76], all measures of gross motor capacity and/or performance. Although many studies also just used single instruments to assess various outcome domains (Table 1). Furthermore, the total number of instruments used per study varied substantially. Whereas Feng 2015 [26] only assessed safety, Min 2020 [41] administered 18 instruments (including safety and biomarker assessments) (Table 1).

#### Descriptive Outcomes

In addition to the above reported outcome instruments, there were numerous descriptive/observational outcomes reported. For instance, 24/54 (44%) studies included purely descriptive outcome/s for at least one outcome category/sub-category (Table 1). Some studies were heavily weighted to reporting descriptive outcomes almost exclusively, particularly case series/reports or single-arm studies. Moreover, some outcome sub-categories were more often reported via descriptive means than an outcome instrument. For example, as mentioned above, although *Executive Function* was assessed using the BRIEF in only one study, it was captured descriptively in another five studies.

Of particular note are the descriptive/observational outcomes classified under the *Other* category. These covered a range of outcomes including sleep, sensory (sensory processing/smell), vision, hearing, appetite and immunity, as well as overarching/comprehensive assessments of participant condition/well-being (Table 1).

### 3.7. Outcome Instrument Properties

The properties of all reported instruments including format, primary purpose and population designed for are shown in Table 2. The largest proportion of instruments (55%) were either exclusively, or partially, Performance-based measures. Clinician-reported measures were the next most commonly utilized, representing 26% of instruments, and these were typically Observations. Other measures were either Parent/other-reported or could be completed interchangeably by a clinician, parent/other or the participant themselves. Only one instrument was exclusively Self-reported (Modified Rankin Scale). In general, across the various outcome sub-/categories, there was a mix between Performance-based and Clinician-reported measures, although *Brain Structure and Function* was exclusively Clinician Observation. The three Quality of Life instruments were all Parent/other or Self-reported, and most *Behavior* and *Activities of Daily Living* assessment tools included input from Parent/other (Table 2).

Of the 53 instruments, 33 (62%) were determined to be evaluative measures, 14 discriminative and/or predictive, and three were classification systems. Of particular note, all instruments within *Brain Structure and Function* were designated as discriminative/predictive, and the *Language and Communication* outcome tools were also primarily non-evaluative. Finally, 14 (26%) of the instruments were specifically designed for a CP-population, mostly within the *Movement and Posture* category. An additional six were designed for adult and/or pediatric rehabilitation and the remainder are for non-specific (general) populations.

When comparing the 53 reported outcome instruments against the highly recommended tools within the common data elements for CP [10], only six instruments overlapped: the GMFM, Tardieu Scale [77], Bayley Scales of Infant and Toddler Development (BSID) [78], Wechsler Intelligence Scale for Children (WISC) [79], BRIEF, and the Cerebral Palsy Quality of Life Questionnaire (CP QOL) [80]. Of note, the GMFCS, MACS and CFCS are also recommended in the common data elements, but as classification systems.

## 4. Discussion

Stem cells and cell therapies offer great potential as a treatment for CP, with efficacy demonstrated in systematic reviews [4,5]. Improvements in gross motor function have been the most commonly studied outcome in randomized controlled trials, however individuals with CP and their families cite improvements in various domains to be of value [7,9]. We conducted this scoping review to describe all outcomes reported in cell therapy studies for CP to date. From this, we wanted to understand whether clinical study outcomes align with common comorbidities and complications of CP, and hence whether they are meeting the expectations of trial participants and their families. Furthermore, we aimed to examine the instruments that are being used to assess these outcomes, to determine whether they are being captured appropriately.

We found that, across 54 included studies comprising >2000 participants, a large range of outcome domains/categories were reported. Notably, *Movement and Posture* was the most commonly assessed outcome category, captured in 98% of included studies. This is understandable given that CP is clinically characterized by motor and postural impairments. Movement and posture are routinely measured within CP clinical studies investigating a whole host of interventions, with several validated instruments with good psychometric properties available for the CP population [81]. *Safety* was the next most common outcome domain. Again, not surprising since clinical studies must necessarily focus on assessing and reporting the safety of experimental intervention/s. Specifically, Phase 1, 2 clinical trials are important for understanding how a drug interacts with the human body, and to identify adverse events. Subsequent Phase 3 clinical trials, including larger numbers of participants, are important to show long-term or rare side effects. Importantly, previous systematic reviews have reported an encouraging safety profile for cell therapy treatments in individuals with CP [4,5], giving confidence to the field in pursuing these novel interventions.

### 4.1. Alignment of Reported Outcomes with Symptoms, Comorbidities and Complications of CP

Some interesting observations were noted when evaluating the reported outcome categories against frequently occurring symptoms, comorbidities and complications of CP [6]. Firstly, whilst many common impairments and functional limitations were captured in the included studies (e.g., walking, talking, epilepsy (seizures), intellect and behavior), the frequency with which these were reported often differed markedly from their prevalence in the CP population. For example, as previously mentioned, gross motor was captured for 86% of participants as expected for a condition defined by limitations to movement and posture. However, other comorbidities/functional limitations with high prevalence in CP were underrepresented. These include intellectual disability (1 in 2 children with CP, but only assessed for 35% of participants), speech impairment (1 in 3 children with CP, but only assessed for 24% of participants), behavior disorders (1 in 4 children with CP, but only assessed for 49% of participants) and epilepsy (seizures) (1 in 4 children with CP, but only assessed for 18% of participants) [6]. In addition, some comorbidities and complications were reported for only a minority of participants using primarily descriptive measures, or not reported at all, despite being commonly occurring, in particular vision impairment (1 in 4 children with CP), pain (3 in 4 children with CP) and sleep disorders (1 in 5 children with CP) [6]. While questions relating to pain and sleep are included in measures of quality of life, these contribute towards the construct of quality of life rather than being assessments of pain or sleep in their own right. *Quality of Life* was captured for only 14% of participants, and of these, more than a third were assessed using health-related-specific quality of life measures. We know that quality of life is influenced by a broad array of factors (i.e., more than health), including socioeconomic status and community life, impacted by social policy such as inclusion, participation, community, and accessibility [82]. Given that quality of life was identified as the most important domain for improvement following intervention via a Delphi survey of youth with CP, parents of children with CP, and medical professionals [9], it is interesting that this was not captured more broadly. We advocate that outcome measures that assess overarching quality of life, with responsiveness to change, such as the CPCHILD [83] for children with severe physical disability [10,84], should be included in future studies.

Important to consider is why many of these prevalent comorbidities, complications and functional limitations are not typically reported in clinical studies of cell therapies to date. Whilst it may be due to a lack of availability or knowledge of suitable/appropriate measurement tools for these outcomes, it is also possible that it is not scientifically plausible for cell therapies to target all of these domains. Indeed, there is some debate in the field as to what potential benefits various cell therapies are actually capable of bestowing [85]. Whilst cell therapies have been under investigation for decades (both clinically and pre-clinically), a comprehensive understanding of the mechanism/s of action for each cell type is still being uncovered. For example, it is accepted that neural stem cells can differentiate into neurons, oligodendrocytes and astrocytes to potentially replace lost or damaged brain cells. On the other hand, cell types including mesenchymal stem/stromal cells and hematopoietic stem cells, which were frequently administered in the studies included in this review, are more ambiguous in their mechanism/s of action for CP [4]. Moreover, how various potential mechanisms of action may relate to the likelihood of improvement across different outcome domains (e.g., gross motor vs. cognition vs. pain) remains unknown. Despite these uncertainties, accumulating high-quality evidence exists to support the efficacy of various cell therapies for improving gross motor function in CP, and there is lower-quality evidence suggesting that cell therapies can have wide-ranging effects across many other domains. This includes various anecdotes and descriptive measures, and while this information can be useful in providing hints at potential areas of efficacy, these subjective reports should be verified using valid tools, in well-designed and powered clinical trials, to determine if they are indeed true effects of a cell treatment. Furthermore, a thorough review of the clinical literature across various conditions that share some of the common comorbidities and complications of CP may help identify additional beneficial effects of cell therapies on these treatment targets.

Another reason why common comorbidities, complications and functional limitations of CP are absent in clinical trials may be a ‘carry-over’ from preclinical (primarily rodent/small animal) research. A known limitation of many animal models is the inadequacy to faithfully replicate the complexity of human disease [86], in addition to difficulties assessing traditionally self-reported outcomes, such as pain [87]. Thus, some outcomes may get overlooked when translating promising cell therapies from the ‘bench’ to the clinic. This highlights the importance of consumer engagement and co-design in medical research, to ensure that research, in particular clinical trials, are informed by community priorities, whilst remaining balanced with what scientists believe, and evidence tells us, cell therapies can feasibly achieve. We therefore recommend that future trials are designed in collaboration with consumer and community representatives to ensure included outcomes are aligned with consumer priorities.

### 4.2. Appropriate Outcome Instrument Selection in Cell Therapy Clinical Studies for CP

Regardless of the outcome domain/s being assessed, it is vitally important that psychometrically sound and appropriate instruments are utilized. This will ensure that data generated from costly and time-consuming clinical trials is high quality and will not lead to incorrect conclusions about the efficacy (or lack thereof) of a particular intervention. This review revealed a large number (>50) of instruments used across the included studies. Encouragingly, many were ‘gold standard’ CP outcome measures, with responsiveness to change, such as the GMFM and the PEDI/PEDI-CAT, which were the two most frequently utilized measures. In contrast, it was concerning that the GMFCS was used to capture change following intervention for a substantial number of participants (the third most frequent outcome tool used). Whilst the GMFCS is a widely used tool for the classification of gross motor function in children with CP, it is not an evaluative measure (i.e., it was not designed, nor shown to be, responsive to change), and is thus not appropriate to be used as an instrument to detect change following an intervention. Interestingly, two other classification tools were also used: the Manual Ability Classification Scale (MACS) [88] and the Communication Function Classification System (CFCS) [89]. We recommend that these classification systems are not used as outcome assessment instruments in future studies.

Excluding the classification tools, two-thirds of all instruments reported had evaluative properties, making them suitable as outcome assessment instruments. Some outcome categories however, were primarily assessed using inappropriate instruments in terms of their evaluative properties, e.g., *Language and Communication*. There are various reasons why inappropriate instruments may be used in clinical trials, including a lack of knowledge, training, or access (e.g., funding). Alternatively, there may as yet be no widely accepted, and validated, evaluative tools for assessing that particular outcome in CP. There are excellent reviews that have identified valid and reliable measures for use in studies of children and youth with CP [81]. However, if suitable tools do not exist, we propose that these areas are not ready for measurement within clinical trials or that individualized goal setting tools might be considered.

Another consideration for selection of outcome domain/s and assessment tools relates to the heterogeneity of CP. Some may argue that the inherent variability between individuals with CP precludes the inclusion and measurement of particular outcomes because they may not be relevant for a large proportion of trial participants, e.g., hearing or vision impairment, or epilepsy. Yet, there is precedent for the use of individualized outcome measures, for example the Goal Attainment Scale (GAS) [90] or Canadian Occupational Performance Measure (COPM) [91] within clinical trials to importantly capture change that matters to the child and family. The use of such measures may enhance the relevancy of captured outcomes for a given participant, help to limit the total number of assessments, thereby reducing respondent burden, and improve sensitivity to detect meaningful change. Thus, it would be interesting to see whether such measures could be used in future trials.

### 4.3. Mechanisms of Cell Therapies and Ensuing Effects

CP is caused by an interference, lesion, or abnormality of the developing brain which manifests as a disorder of movement and/or posture. Repairing the underlying brain injury, via direct or indirect mechanisms, to promote increased neuronal signaling and function is the aim of cell therapies for CP. As such, it is recognized that improvements in brain structure or connectivity following cell intervention could directly improve motor function. It is important to acknowledge however that links exist between motor skills and some comorbidities of CP. Figure 3 shows a schema of the proposed effects of stem cells for CP including therapeutic targets leading to remediation of the underlying brain injury, and resultant effects on various comorbidities, leading to the ultimate goal of improving quality of life. We wish to specifically highlight that changes in brain structure and connectivity producing improvements in motor function may have secondary effects on a number of motor-associated CP comorbidities (e.g., pain, sleep, drooling and speech). This may therefore mean that, in fact, improvements in various outcomes of importance to individuals with CP and their families may be more achievable than widely believed. In addition, the non-motor-associated comorbidities of CP (e.g., cognition, behavior) may be indirectly targeted by cell treatments.

### 4.4. Limitations

We acknowledge some limitations of this scoping review including that due to our decision to include all study designs, there is a significant amount of lower-quality evidence included. In addition, extracted outcome instruments may have been categorized in varying ways, and, for simplicity of reporting, some sub-categories of outcomes were consolidated during the sorting process, despite arguably representing distinct outcome sub-domains. Finally, we did not extract nor report on the efficacy of cell therapies for any of the outcome categories, as this was outside the scope of this review.

## 5. Conclusions

Stem cells are an emerging intervention for CP with potential to target a wide variety of outcome domains. We found that movement and posture and safety were the predominant outcomes assessed in cell therapy clinical studies, despite many other outcomes, including quality of life, being of high importance to individuals with CP and their families. Moreover, amongst the considerable number of outcome instruments employed in clinical studies, many are not appropriate for use as measures of change following intervention. We provide several recommendations to ensure that future trials collect scientifically valid, high-quality outcome data that also meets the expectations of the CP community.

## Figures and Tables

**Figure 1 jcm-11-07319-f001:**
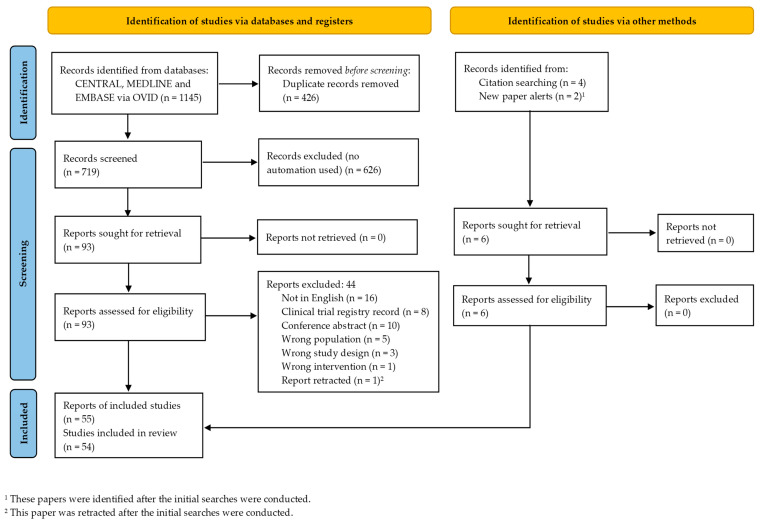
PRISMA flow diagram of study selection.

**Figure 2 jcm-11-07319-f002:**
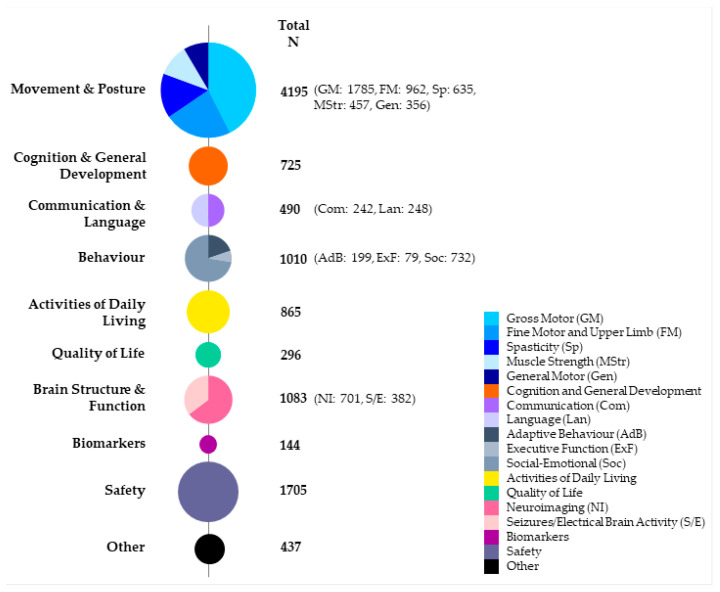
Number of participants assessed for each outcome category and sub-category across all included studies. N, number of participants.

**Figure 3 jcm-11-07319-f003:**
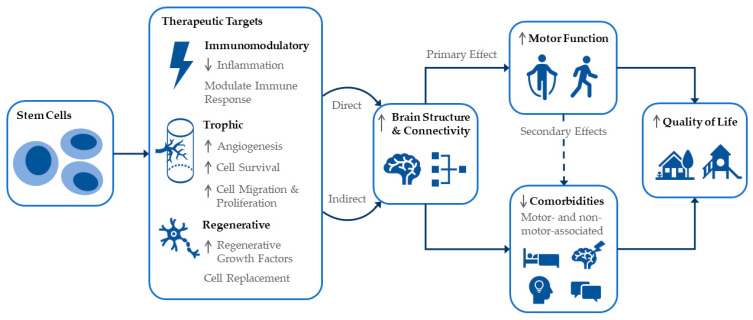
Schematic representation of cell intervention effects and interlinked outcomes for CP including quality of life.

**Table 1 jcm-11-07319-t001:** Details of included studies.

Study Reference	Study Design	Participant Details at Baseline	Intervention/s and Comparator/s	Cell Therapy, Donor Type and Route	Last Follow Up Post-Cell Treatment ^1^	Outcome Sub-Categories: Instrument/s Reported	Level of Evidence ^2^
AbiChahine 2016 [16]	Case series n = 17 n = 2 LTFU	Subtype: Various Severity: GMFCS I-II, IV-V Age: 1.5–17 years	BM-MNCs (n = 17)	Autologous, intrathecal	Not reported	**Gross Motor:** GMFCS **Spasticity; Cognition & General Development; Activities of Daily Living; Adaptive Behavior**: Descriptive **Safety**	4
Amanat 2021 [17] ^3^	RCT n = 72 n = 5 LTFU	Subtype: Spastic quadriplegia and diplegia Severity: GMFCS II-V Age: Mean 8.5 years	Group 1: UC-MSCs + rehab (n = 36) Group 2: Sham procedure + rehab (n = 36)	Allogeneic, intrathecal	1 year	**Gross Motor:** GMFM-66, PEDI **Spasticity:** MAS **Activities of Daily Living:** PEDI **Social-Emotional:** PEDI **Quality of Life:** CP-QoL **Neuroimaging:** MRI-DTI **Safety**	2
Bansal 2016 [18]	Single-arm n = 10	Subtype: Not reported Severity: GMFCS II-IV Age: 2–10 years	BM-MNCs + rehab (n = 10)	Autologous, intrathecal	2 years	**Gross Motor:** GMFCS **Fine Motor & Upper Limb:** MACS **Communication:** CFCS **Spasticity:** Descriptive **Neuroimaging:** MRI **Safety**	4
Boruczkowski 2019 [19]	Case series (Retrospective) n = 107 n = 17 LTFU + n = 36–67 missing data (outcome dependent)	Subtype: Not reported Severity: Not reported Age: 1.4–17 years	UC-MSCs (n = 107)	Allogeneic, intravenous	Not reported	**Gross Motor; Fine Motor & Upper Limb; Spasticity; Muscle Strength; Quality of Life; Activities of Daily Living; Cognition & General Development; Adaptive Behavior; Executive Function; Social-Emotional; Communication; Seizures/Electrical Brain Activity:** Descriptive **Other:** Descriptive (sensory, sleep, circulation, medications) **Safety**	4
Chen 2010 [20]	RCT n = 33 n = 19 LTFU	Subtype: Not reported Severity: Not reported Age: 1–12 years	Group 1: Fetal OECs + rehab (n = 18) Group 2: Rehab alone (n = 15)	Allogeneic, intracerebral	6 months	**Gross Motor:** GMFM-66 **Other:** Caregiver Questionnaire Scale **Safety**	2
Chen 2013 [21]	Non-randomised controlled n = 60	Subtype: Not reported Severity: GMFCS III-V Age: 1–35 years	Group 1: BM-MSC-derived NSC-like cells + rehab (n = 30) Group 2: Rehab alone (n = 30)	Autologous, intrathecal	6 months	**Gross Motor:** GMFM-88 **Language:** Gesell Developmental Schedules **Safety**	3
Chernykh 2014 [22]	Single-arm n = 21	Subtype: Various Severity: GMFCS IV-V Age: 2–8 years	Peripheral blood expanded M2-like macrophages (n = 21)	Autologous, intrathecal	5 years	**Gross Motor:** GMFM-66 **Fine Motor & Upper Limb:** PDMS-FM **Spasticity:** Ashworth Scale **Muscle Strength:** MRC Scale **Cognition & General Development; Seizures/Electrical Brain Activity:** Descriptive **Other:** Descriptive (infections, temperatures) **Biomarkers** **Safety**	4
Chernykh 2018 [23]	Single-arm n = 57	Subtype: Various Severity: GMFCS III-V Age: 1–10 years	Peripheral blood expanded M2-like macrophages (n = 57)	Autologous, intrathecal	5 years	**Gross Motor:** GMFM-66 **Fine Motor & Upper Limb:** PDMS-FM **Spasticity:** Ashworth Scale **Muscle Strength:** MRC Scale **Cognition & General Development; Seizures/Electrical Brain Activity:** Descriptive **Biomarkers** **Safety**	4
Cox 2022 [66]	RCT: Cross-over n = 20 n = 2 LTFU (longer term endpoint only)	Subtype: Various Severity: GMFCS II-V Age: 2.4–10.9 years	Group 1: UCB then placebo (n = 3) Group 2: BM-MNCs then placebo (n = 10) Group 3: Placebo then UCB (n = 2) Group 4: Placebo then BM-MNCs (n = 5)	Autologous, intravenous	2 years (1 year post cross-over)	**Gross Motor:** GMFM-66/-88 **General Motor:** VABS-2 **Communication:** VABS-2 **Activities of Daily Living:** VABS-2 **Adaptive Behavior:** VABS-2 **Social-Emotional:** VABS-2 **Neuroimaging:** MRI/MRI-DTI **Safety**	2
Crompton 2022 [24]	Single-arm n = 12 ^4^ n = 1 withdrew before treatment	Subtype: Various Severity: GMFCS I-V Age: 2.7–11.6 years	UCB (n = 12)	Allogeneic, intravenous	1 year	**Gross Motor:** GMFM-66 **Fine Motor & Upper Limb:** QUEST **General Motor:** VABS-2 **Communication:** VABS-2 **Activities of Daily Living:** VABS-2 **Cognition & General Development:** BSID-3, WPPSI-IV or WISC-V **Adaptive Behavior:** VABS-2 **Executive Function:** BRIEF **Social-Emotional:** SDQ, VABS-2 **Quality of Life:** CP-QoL-Child **Safety**	4
Dong 2018 [25]	Case report n = 1	Subtype: Not reported Severity: Not reported Age: 4 years	UC-MSCs (n = 1)	Donor type not specified, intravenous and intrathecal	Not reported	**Seizures/Electrical Brain Activity:** EEG **Muscle Strength; General Motor; Language; Cognition & General Development:** Descriptive	4
Feng 2015 [26]	Case series (Retrospective) n = 47	Subtype: Not reported Severity: “Severe” Age: 1–29 years	UCB (n = 47)	Allogeneic, intravenous then intrathecal	6 months	**Safety**	4
Fu 2019 [27]	Non-randomised dose comparison n = 60 n = 3 LTFU	Subtype: Spastic, topography not reported Severity: GMFCS IV-V Age: Not reported	Group 1: UC-MSCs 1 course (n = 30) Group 2: UC-MSCs 2 courses (n = 27)	Allogeneic, intrathecal	1 year	**Gross Motor:** GMFM-88 **Fine Motor & Upper Limb:** FMFM **Safety**	4
Gabr 2015 [67]	RCT n = 100 n = 6 withdrew before treatment	Subtype: Various Severity: GMFCS II-V Age: Mean 4.8 years	Group 1: BM-MSCs (n = 44) Group 2: Standard care (n = 50)	Autologous, intrathecal	1 year	**Gross Motor:** GMFCS, PEDI **Quality of Life:** CHQ **Activities of Daily Living:** PEDI **Social-Emotional:** PEDI **Safety**	2
Gu 2020 [28]	RCT n = 40 n = 1 withdrew before treatment	Subtype: Not reported Severity: Not reported Age: Mean 4.3 years	Group 1: UC-MSCs + rehab (n = 19) Group 2: Placebo + rehab (n = 20)	Allogeneic, intravenous	1 year	**Gross Motor:** GMFM-88 **Activities of Daily Living:** ADL **Neuroimaging:** PET-CT **Other:** CFA **Safety**	2
Hassan 2012 [29]	Non-randomised controlled n = 52	Subtype: Athetoid and spastic, various topography Severity: GMFCS unclear ^5^ Age: 1–8 years	Group 1: BM-MSC (n = 26) Group 2: No treatment (n = 26)	Autologous, intrathecal	1 year	**Gross Motor:** GMFCS, BDPS **Activities of Daily Living:** BDPS **Communication:** BDPS **Other:** Descriptive (‘100 points scale’)	3
Hirano 2018 [68]	Case report n = 1	Subtype: Hemiplegia, type not reported Severity: GMFCS II Age: 7 years	Adipose-MSCs (n = 1)	Allogeneic, intravenous, intramuscular, subcutaneous and intra-articular	1 year	**Gross Motor:** GMFCS **Quality of Life:** SF-8 **Other:** Descriptive (clinical condition) **Safety**	4
Huang 2018 [31]	RCT n = 56 n = 2 LTFU	Subtype: Not reported Severity: Not reported Age: 3–12 years	Group 1: UCB-MSCs + rehab (n = 27) Group 2: Placebo + rehab (n = 27)	Allogeneic, intravenous	2 years	**Gross Motor:** GMFM-88 **Neuroimaging:** MRI **Seizures/Electrical Brain Activity:** EEG **Other:** CFA **Safety**	2
Jensen 2016 [32]	Case report n = 1	Subtype: Spastic hemiplegia Severity: GMFCS I equivalent Age: 5 years	UCB + rehab (n = 1)	Autologous, intravenous	5.5 years	**Gross Motor; Fine Motor & Upper Limb; Spasticity; Muscle Strength; Cognition & General Development:** Descriptive **Safety**	4
Kang 2015 [33]	RCT n = 36 n = 2 withdrew before treatment	Subtype: Not reported Severity: GMFCS I-V Age: 0.5–18 years	Group 1: UCB + rehab (n = 17) Group 2: Placebo + rehab (n = 17)	Allogeneic, intravenous or intra-arterial	6 months	**Gross Motor:** GMFM, GMPM, WeeFIM, PEDI **Muscle Strength:** MMT score **General Motor:** BSID-2 ^6^ **Cognition & General Development:** WeeFIM **Activities of Daily Living:** WeeFIM, PEDI **Social-Emotional:** PEDI **Neuroimaging:** PET-CT **Biomarkers** **Safety**	2
Kikuchi 2022 [70]	Single-arm n = 6	Subtype: Spastic hemiplegia, diplegia and quadriplegia Severity: GMFCS I, III-V Age: 1.7–6.7 years	UCB (n = 6)	Autologous, intravenous	3 years	**Gross Motor:** GMFM-66, GMFCS **General Motor:** KSPD **Cognition & General Development:** KSPD, WISC-IV **Social-emotional:** KSPD **Neuroimaging:** MRI-DTI **Seizures/Electrical Brain Activity:** EEG **Safety**	4
Lee 2012 [34]	Single-arm n = 20	Subtype: Various topography, type not reported Severity: Not reported Age: 1.9–7.6 years	UCB (n = 20)	Autologous, intravenous	6 months	**Gross Motor:** GMFM-88, GMFCS, PEDI, DDST-2 **Fine Motor & Upper Limb:** QUEST, MACS, DDST-2 **Activities of Daily Living:** PEDI **Social-Emotional:** PEDI, DDST-2 **Language:** DDST-2 **Neuroimaging:** MRI-DTI, SPECT	4
Li 2012 [35]	Case report n = 1	Subtype: Not reported Severity: Ambulant Age: 11 years	BM-MSCs (n = 1)	Autologous, intravenous	1 year	**Spasticity:** Descriptive **Other:** Descriptive (vision) **Safety**	4
Liu 2017 [36]	RCT n = 105 n = 3 LTFU	Subtype: Spastic, topography not reported Severity: GMFCS II-V Age: 0.5–12.5 years	Group 1: BM-MSCs (n = 35) Group 2: BM-MNCs (n = 35) Group 3: Rehab (n = 35)	Autologous, intrathecal	1 year	**Gross Motor:** GMFM **Fine Motor & Upper Limb:** FMFM	2
Luan 2012 [37]	RCT n = 94	Subtype: Various Severity: “Severe” Age: Mean 1.3 years	Group 1: Fetal NPCs + rehab (n = 45) Group 2: Rehab alone (n = 49)	Allogeneic, intra-cerebroventricular	1 year	**Gross Motor:** GMFM **Fine Motor & Upper Limb:** PDMS-FM **Cognition & General Development:** Descriptive **Other:** Descriptive (sleep) **Safety**	2
Mancias-Guerra 2014 [38]	Single-arm n = 18 n = 5 LTFU	Subtype: Various Severity: Not reported Age: 2.2–5.5 years	BM-TNCs (n = 18)	Autologous, intrathecal and intravenous	6 months	**General Motor:** BDI **Cognition & General Development:** BDI **Communication:** BDI **Adaptive Behavior:** BDI **Social-Emotional:** BDI **Neuroimaging:** MRI **Safety**	4
Maric 2022 [39]	Single-arm n = 42	Subtype: Various types, topography not reported Severity: GMFCS I-V Age: 1–12 years	BM-MNCs (n = 42)	Autologous, intrathecal	1 year	**Gross Motor:** GMFCS, S-D **Fine Motor & Upper Limb:** LAP-D **Spasticity:** MAS **Cognition & General Development:** LAP-D **Language:** LAP-D **Neuroimaging:** MRI **Seizures/Electrical Brain Activity:** EEG **Safety**	4
Min 2013 [40]	RCT n = 105 n = 9 LTFU	Subtype: Various Severity: GMFCS I-V Age: 0.6–9.8 years	Group 1: UCB + EPO + rehab (n = 35) Group 2: Placebo UCB + EPO + rehab (n = 36) Group 3: Placebo UCB + Placebo EPO + rehab (n = 34)	Allogeneic, intravenous	6 months	**Gross Motor:** GMFM, GMPM, PEDI, WeeFIM **Fine Motor & Upper Limb:** QUEST **Muscle Strength:** MMST **General Motor:** BSID-2 **Cognition & General Development:** BSID-2, WeeFIM **Activities of Daily Living:** PEDI, WeeFIM **Social-Emotional:** PEDI **Neuroimaging:** MRI-DTI, PET-CT **Safety**	2
Min 2020 [41]	RCT n = 92 n = 4 LTFU	Subtype: Various Severity: GMFCS I-V Age: 1–6.3 years	Group 1: UCB + EPO (n = 22) Group 2: UCB + Placebo EPO (n = 24) Group 3: Placebo UCB + EPO (n = 20) Group 4: Placebo UCB + Placebo EPO (n = 24)	Allogeneic, intravenous	1 year	**Gross Motor:** GMFM, GMPM, GMFCS, PEDI, SCALE **Fine Motor & Upper Limb:** QUEST **Spasticity:** MAS, Modified Tardieu Scale **Muscle Strength:** MRC Scale **General Motor:** BSID-2 **Cognition & General Development:** BSID-2, FIM **Activities of Daily Living:** FIM, PEDI **Social-Emotional:** PEDI **Neuroimaging:** MRI-DTI, PET-CT **Seizures/Electrical Brain Activity:** EEG **Other:** Descriptive (parent satisfaction), Beery VMI **Biomarkers** **Safety**	2
Nguyen 2017 [42]	Single-arm n = 40	Subtype: Spastic bilateral and unilateral Severity: GMFCS III-V Age: 1–12 years	BM-MNCs (n = 40)	Autologous, intrathecal	6 months	**Gross Motor:** GMFM-66/-88 **Spasticity:** MAS **Safety**	4
Nguyen 2018 [43]	Single-arm n = 30	Subtype: Quadriplegia and hemiplegia, type not reported Severity: GMFCS II-V Age: 2–15.5 years	BM-MNCs + rehab (n = 30)	Autologous, intrathecal	6 months	**Gross Motor:** GMFM-66/-88 **Spasticity:** MAS **Quality of Life:** CP-QoL-Child	4
Okur 2018 [44]	Case report n = 1	Subtype: Dystonic Severity: GMFCS V Age: 6 years	UC-MSCs + rehab (n = 1)	Allogeneic, intrathecal, intramuscular and intravenous	1.5 years	**Gross Motor:** GMFCS, TCMS **Fine Motor & Upper Limb:** MACS **Spasticity:** Modified Tardieu Scale **Communication:** CFCS **Cognition & General Development:** FIM **Activities of Daily Living:** FIM **Safety**	4
Padma Srivastava 2011 [45]	Case series n = 30	Subtype: Dystonic and spastic ^7^, topography not reported Severity: “Moderate to severe” Age: 5–25 years	BM-MNCs (n = 30)	Autologous, intra-arterial	1 year	**Spasticity:** Ashworth Scale **Muscle Strength:** MRC Scale **Activities of Daily Living:** mBI **Other:** mRS **Safety**	4
Papadopoulos 2011 [46]	Case report n = 2	Subtype: Spastic diplegia Severity: GMFCS III Age: 1.6 and 2.7 years	Case 1: UCB + G-CSF 12 months post-infusion Case 2: UCB + G-CSF pre- and post-infusion	Autologous, intravenous	**Case 1:** 2.3 years **Case 2:** 7 months	**Gross Motor:** GMFCS **Neuroimaging:** MRI **Spasticity:** Descriptive **Safety**	4
Purandare 2012 [47]	Case report n = 1	Subtype: Not reported Severity: GMFCS III Age: 6 years	BM-MNCs (n = 1)	Autologous, intrathecal	2 years	**Gross Motor:** GMFCS **Neuroimaging:** PET-CT **Seizures/Electrical Brain Activity:** EEG **Fine Motor & Upper Limb; Cognition & General Development; Executive Function; Language:** Descriptive **Other:** Descriptive (sensory)	4
Purwati 2019 [48]	Single-arm n = 14 n = 2 LTFU ^8^	Subtype: Not reported Severity: GMFCS III-IV Age: 1–11 years	Adipose-derived NPCs (n = 12)	Autologous, intra-cerebroventricular	1 year	**Gross Motor:** GMFCS **Spasticity; Cognition & General Development; Communication:** Descriptive **Safety**	4
Rah 2017 [49]	RCT: Cross-over n = 57 n = 10 LTFU	Subtype: Various Severity: “Non-severe” Age: 2–10 years	Group 1: Peripheral blood-MNCs then placebo (n = 28) Group 2: Placebo then peripheral blood-MNCs (n = 29)	Autologous, intravenous	1 year (6 months post cross-over)	**Gross Motor:** GMFM-88, GMFCS, PEDI, DDST-2 ^9^ **Fine Motor & Upper Limb:** MACS, QUEST **Activities of Daily Living:** PEDI **Social-Emotional:** PEDI **Neuroimaging:** MRI-DTI, PET-CT **General Motor; Cognition & General Development**: Descriptive **Safety**	2
Ramirez 2006 [69]	Single-arm n = 8	Subtype: Various types, topography not reported Severity: Not reported Age: 3–12 years	Expanded UCB CD133+ cells (n = 8)	Allogeneic, subcutaneous intramuscular	6 months	**Gross Motor; Fine Motor & Upper Limb; Spasticity; Cognition & General Development; Communication; Language:** Descriptive **Other:** Descriptive (infections, vision) **Safety**	4
Romanov 2015 [50]	Case series (Retrospective) n = 80 n = 25 LTFU/excluded + n = 17–19 missing data (outcome dependent)	Subtype: Various Severity: GMFCS IV-V Age: 1–12 years	UCB (n = 80)	Allogeneic, intravenous	3 years post first treatment	**Gross Motor:** GMFCS **Spasticity:** MAS **Muscle strength:** Hand dynamometry **Safety**	4
Seledtsov 2005 [51]	Non-randomised controlled n = 60	Subtype: Double hemiplegia, spastic diplegia and atonic-astatia Severity: “Severe” Age: 1.5–7 years	Group 1: Fetal nervous and hematopoietic cells (n = 30) Group 2: Standard care (n = 30)	Allogeneic, intrathecal	1 year	**Seizures/Electrical Brain Activity:** EEG **Gross Motor; Fine Motor & Upper Limb; Cognition & General Development; Communication; Language:** Descriptive **Other:** Descriptive (‘100 points scale’, vision) **Safety**	3
Sharma 2013 [52]	Case report n = 1	Subtype: Spastic diplegia Severity: GMFCS III equivalent Age: 20 years	BM-MNC + rehab (n = 1)	Autologous, intrathecal	1 year	**Cognition & General Development:** FIM, IQ Score **Activities of Daily Living:** FIM **Neuroimaging:** PET-CT **Other:** Mental Status Examination, Descriptive (appetite) **Gross Motor; Fine Motor & Upper Limb; Executive Function, Social-Emotional; Communication; Language:** Descriptive	4
Sharma 2015 [53]	Single-arm n = 40	Subtype: Various Severity: GMFCS I-V Age: 1.4–22 years	BM-MNCs + rehab (n = 40)	Autologous, intrathecal	6 months	**Neuroimaging:** PET-CT **Gross Motor; Fine Motor & Upper Limb; Spasticity; Muscle Strength; General Motor; Activities of Daily Living; Cognition & General Development; Social-Emotional; Language:** Descriptive **Safety**	4
Sharma 2015 [54]	Case report n = 1	Subtype: Spastic diplegia Severity: GMFCS III Age: 12 years	BM-MNCs + rehab (n = 1)	Autologous, intrathecal	1 year	**Cognition & General Development:** FIM **Activities of Daily Living:** FIM **Neuroimaging:** PET-CT **Gross Motor; Fine Motor & Upper Limb; Muscle Strength:** Descriptive **Other:** Descriptive (sense of smell)	4
Sharma 2020 [55]	Case report n = 1	Subtype: Spastic diplegia Severity: GMFCS III Age: 4 years	BM-MNCs + rehab (n = 1)	Autologous, intrathecal	1.3 years post first treatment	**Gross Motor:** GMFM, GMFCS **Cognition & General Development:** FIM **Activities of Daily Living:** FIM **Neuroimaging:** PET-CT **Fine Motor & Upper Limb; Spasticity; Muscle Strength; General Motor; Adaptive Behavior; Executive Function:** Descriptive **Other:** Descriptive (sensory processing) **Safety**	4
Shroff 2014 [56] ^10^	Case series (Retrospective) n = 101 n = 10 excluded from analysis n= 25−76 LTFU between treatment phases	Subtype: Not reported Severity: GMFCS I-V Age: ≤2 to 18 years	ESCs + rehab (n = 101)	Allogeneic, multiple routes ^11^	2.4 years post first treatment	**Gross Motor:** GMFCS **Neuroimaging:** SPECT **Activities of Daily Living; Cognition & General Development; Executive Function; Social-Emotional; Language; Seizures/Electrical Brain Activity:** Descriptive **Other:** Descriptive (hearing) **Safety**	4
Sun 2017 [57]	RCT: Cross-over n = 63	Subtype: Various Severity: GMFCS I-IV Age: 1.1–7 years	Group 1: UCB then placebo (n = 32) Group 2: Placebo then UCB (n = 31)	Autologous, intravenous	2 years (1 year post cross-over)	**Gross Motor:** GMFM-66, PDMS **Fine Motor & Upper Limb:** PDMS **Neuroimaging:** MRI-DTI **Safety**	2
Sun 2021 [58]	Single-arm n = 15	Subtype: Spastic, various topography Severity: GMFCS II-IV Age: 1–6 years	UCB (n = 15)	Allogeneic, intravenous	2 years	**Gross Motor:** GMFM-66, PDMS **Fine Motor & Upper Limb:** AHA, PDMS **Safety**	4
Sun 2022 [71]	RCT n = 91 n = 1 withdrew before treatment + n = 22 LTFU incl. 18 due to COVID-19	Subtype: Hypertonic, various topography Severity: GMFCS I-IV Age: 2.1–5 years	Group 1: UCB (n = 31) Group 2: UC-MSCs (n = 28) Group 3: Control (n = 31)	Allogeneic, intravenous	1 year	**Gross Motor:** GMFM-66, PDMS, PEDI-CAT **Fine Motor & Upper Limb:** PDMS **Activities of Daily Living:** PEDI-CAT **Adaptive Behavior:** PEDI-CAT **Social-Emotional:** PEDI-CAT **Safety**	2
Thanh 2019 [59]	Single-arm n = 25	Subtype: Spastic bilateral Severity: GMFCS II-V Age: 2–15 years	BM-MNCs + rehab (n = 25)	Autologous, intrathecal	1 year	**Gross Motor:** GMFM-66/-88 **Spasticity:** MAS	4
Wang 2013 [60]	Case report n = 1	Subtype: Not reported Severity: Not reported Age: 5 years	UC-MSCs + rehab (n = 1)	Allogeneic, intravenous and intrathecal	2.3 years	**Cognition & General Development:** FIM **Activities of Daily Living:** FIM **Muscle Strength; Communication:** Descriptive **Other:** Descriptive (immunity)	4
Wang 2013 [61]	Single-arm n = 52 n = 6 withdrew before treatment + n = 6 LTFU	Subtype: Spastic and/or athetoid, topography not reported Severity: GMFCS I-V Age: 0.5–15 years	BM-MSC (n = 46)	Autologous, intrathecal +/− intra-parenchymal	1.5 years	**Gross Motor:** GMFM-66/-88 **Safety**	4
Wang 2015 [62]	Single-arm n = 16	Subtype: Spastic, topography not reported Severity: Not reported Age: 3–12 years	UC-MSC (n = 16)	Allogeneic, intrathecal	6 months	**Gross Motor:** GMFM-88 **Fine Motor & Upper Limb:** FMFM	4
Zali 2015 [63]	Single-arm n = 13 n = 1 LTFU	Subtype: Various types, topography not reported Severity: GMFCS III-V Age: 4–10 years	BM-CD133+ cells (n = 13)	Autologous, intrathecal	6 months	**Gross Motor:** GMFM-66, GMFCS, BBS **Spasticity:** MAS **Activities of Daily Living:** UK FIM + FAM **Cognition & General Development:** UK FIM + FAM **Seizures/Electrical Brain Activity:** EEG **Safety**	4
Zarrabi 2022 [64] ^3^	RCT n = 72 n = 6–9 LTFU (outcome dependent)	Subtype: Spastic quadriplegia and diplegia Severity: GMFCS II-V Age: Mean 9 years	Group 1: UCB + rehab (n = 36) Group 2: Sham procedure + rehab (n = 36)	Allogeneic, intrathecal	1 year	**Gross Motor:** GMFM-66, PEDI **Spasticity:** MAS **Activities of Daily Living:** PEDI **Social-Emotional:** PEDI **Quality of Life:** CP-QoL **Neuroimaging:** MRI-DTI **Safety**	2
Zhang 2015 [65]	Case report n = 1	Subtype: Not reported Severity: Not reported Age: 0.5 years	UCB-MSCs + rehab (n = 1)	Allogeneic, intravenous	5 years	**Gross Motor:** GMFM-88 **Spasticity:** Ashworth Scale **Neuroimaging:** MRI **Seizures/Electrical Brain Activity:** EEG **Other:** CDCC Infant Mental Development Scale, CFA **Safety**	4

Bolded text denotes outcome sub-categories. Abbreviations: ADL, Activities of Daily Living assessment; AHA, Assisting Hand Assessment; BBS, Berg Balance Scale; BDI, Battelle Developmental Inventory; BDPS, Boyd’s Developmental Progress Scale; Beery VMI, Beery-Buktenica Developmental Test of Visual-Motor Integration 6th edition; BM, bone marrow; BRIEF, Behavior Rating Inventory of Executive Function; BSID-2, Bayley Scales of Infant and Toddler Development 2nd edition; BSID-3, Bayley Scales of Infant and Toddler Development 3rd edition cognitive scale; CDCC, Child Development Center of China; CFA, Comprehensive Functional Assessment; CFCS, Communication Function Classification System; CHQ, Child Health Questionnaire Parent Form 50; COVID-19, coronavirus disease pandemic; CP QOL-Child, Cerebral Palsy Quality of Life Questionnaire for Children; DDST-2, Denver Development Screening Test 2nd edition; DTI, diffusion tensor imaging; EEG, electroencephalogram; ESC, embryonic stem cell; Ex-UCB, expanded umbilical cord blood cells; FIM, Functional Independence Measure; FMFM, Fine Motor Function Measure; G-CSF, granulocyte-colony stimulating factor; GMFCS, Gross Motor Function Classification System; GMFM-66/-88, Gross Motor Function Measure-66 or -88; GMPM, Gross Motor Performance Measure; KSPD, Kyoto Scale of Psychological Development; LAP-D, Learning Accomplishment System Diagnostic Score; LTFU, lost to follow-up; MACS, Manual Ability Classification Scale; MAS, Modified Ashworth Scale; mBI, Modified Barthel Index; MMST, Manual Muscle Strength Test; MMT score, Manual Muscle Testing score; MNC, mononuclear cells; MRC Scale, Medical Research Council Scale; MRI, Magnetic resonance imaging; mRS, Modified Rankin Scale; MSC, mesenchymal stem/stromal cell; n, number of participants; NPC, neural progenitor cell; NSC, neural stem cell; OEC, olfactory ensheathing cell; PB, peripheral blood; PDMS, Peabody Developmental Motor Scales (2nd edition); PDMS-FM, PDMS Fine Motor Test/Quotient; PEDI/-CAT, Pediatric Evaluation of Disability Inventory (Computer Adaptive Test); PET-CT, positron emission tomography and computed tomography scan; QUEST, Quality of Upper Extremity Skills Test; RCT, randomized controlled trial; rehab, rehabilitation; SCALE, Selective Control Assessment of Lower Extremity; SDQ, Strengths and Difficulties Questionnaire; SF-8, Short Form 8 (SF-8) Health Survey Quality of Life questionnaire; SPECT, single photon emission computed tomography scan; TCMS, Trunk Control Measurement Scale; TNC, total nucleated cell; UCB, umbilical cord blood; UK FIM + FAM, UK Functional Independence Measure and Functional Assessment Measure; VABS-2, Vineland Adaptive Behavior Scales 2nd edition; WeeFIM, Functional Independence Measure for Children; WISC-IV/V, Wechsler Intelligence Scale for Children 4th/5th edition; WPPSI-IV, Wechsler Preschool & Primary Scale of Intelligence 4th edition. ^1^ Most studies captured outcomes at multiple timepoints and not all outcomes were assessed at this timepoint. Some studies that administered multiple cell doses calculated follow-up from after the last cell administration. ^2^ Level according to The Oxford Levels of Evidence 2 [12]. ^3^ These are two reports of the same clinical trial (NCT03795974) and share a control group (sham procedure + rehab). ^4^ Participant that withdrew was replaced, so total number of recruited participants was 13. ^5^ GMFCS reported as mean and standard deviation. ^6^ BSID-2 used for motor only. ^7^ Ascertained from text. ^8^ Timing of LTFU (before/after treatment) not clear. ^9^ DDST-2 used for gross motor only. ^10^ An editorial expression of concern was raised in September 2017 regarding the ethics of this study and the potential association of the risk of teratoma formation with the transplantation of embryonic stem cells [73]. ^11^ Routes included intravenous, intrathecal and intramuscular in addition to eye drops, nasal spray, oral drops, ear drops, deep spinal muscle injections and retro bulbar injections, according to the participant’s clinical characteristics.

**Table 2 jcm-11-07319-t002:** Outcome instruments used in cerebral palsy cell therapy studies with details.

Outcome Sub-Category	Instrument [*Subdomain*] ^1^	n ^2^	Format	Primary Purpose	Population Designed for
**Movement & Posture**					
Gross Motor	Gross Motor Function Measure (GMFM) -66/-88	1163	Performance-based	E	CP
	Pediatric Evaluation of Disability Inventory (PEDI)/PEDI-Computer Adaptive Test (CAT) [*Mobility*]	573	Clinician Observation +/− Parent/other Interview	E, D	General
	Gross Motor Function Classification System (GMFCS)	533	Clinician Observation (or Parent/other/Self Interview)	C	CP
	Gross Motor Performance Measure (GMPM)	218	Performance-based	E	CP
	Peabody Developmental Motor Scales-2 (PDMS-2) [*Gross Motor Quotient*]	132	Performance-based	D (2nd E)	General
	The Functional Independence Measure for Children (WeeFIM) [*Mobility*]	130	Clinician Observation +/− Parent/other Interview	E	Pediatric Rehab
	Selective Control Assessment of Lower Extremity (SCALE)	88	Performance-based	D, E	CP
	Denver Developmental Screening Test 2 (DDST-II) [*Gross Motor*]	67	Performance-based +/− Parent/other Interview	D	General
	Boyd Developmental Progress Scale (BDPS) [*Motor*]	52	Performance-based + Clinician Observation +/− Parent/other Interview	D	General
	Learning accomplishment system diagnostic (LAP-D) Score [*Sitting and Standing*]	42	Performance-based/Parent/other Observation	D	General
	Berg Balance Scale (BBS)	12	Performance-based	P (2nd D, E)	Adult Rehab
	Trunk Control Measurement Scale (TCMS)	1	Performance-based	D, E	CP
Fine Motor and Upper Limb	PDMS-2 [*Fine Motor Quotient*]	304	Performance-based	D (2nd E)	General
	Quality of Upper Extremity Skills Test (QUEST)	262	Performance-based	E	CP (Spastic)
	Fine Motor Function Measure (FMFM)	176	Performance-based	E	CP
	Manual Ability Classification Scale (MACS)	78	Clinician Observation (or Parent/other/Self Interview)	C	CP
	LAP-D [*Fine Motor Skills*]	42	Performance-based/Parent/other Observation	D	General
	DDST-II [*Fine Motor*-*Adaptive*]	20	Performance-based +/− Parent/other Interview	D	General
	Assisting Hand Assessment (AHA)	15	Performance-based	E	CP (Hemiplegia)
Muscle Strength	Medical Research Council (MRC) Scale for Muscle Strength; MRC Summed Scores	196	Performance-based	D, E	General
	Manual Muscle Strength Test	96	Performance-based	D, E	General
	Manual Muscle Testing (MMT) Score	34	Performance-based	D, E	General
	Hand dynamometry	15	Performance-based	D, E	General
Spasticity	Modified Ashworth Scale	381	Performance-based	D, E	CP (Spastic)
	Ashworth Scale	109	Performance-based	D, E	CP (Spastic)
	Modified Tardieu Scale	89	Performance-based	D, E	CP (Spastic)
General Motor ^3^	Bayley Scales of Infant and Toddler Development 2nd Edition (BSID-II) [*Motor Scale*]	218	Performance-based	D (2nd E)	General
	Vineland Adaptive Behavior Scales 2nd Edition (VABS-2), parent report questionnaire [*Motor Skills Domain*]	30	Parent/other Questionnaire	D (2nd P, E)	General
	Battelle Developmental Inventory (BDI) [*Motor*]	13	Performance-based +/− Parent/other Observation/Interview	D (2nd P, E)	General
	Kyoto Scale of Psychological Development (KSPD) [*Postural-Motor*]	6	Performance-based	D, E	General
**Activities of Daily Living**					
Activities of Daily Living	PEDI/PEDI-CAT [*Self-care/Daily Activities*]	573	Clinician Observation +/− Parent/other Interview	E, D	General
	WeeFIM [*Self Care*]	130	Clinician Observation +/− Parent/other Interview	E	Pediatric Rehab
	Functional Independence Measure (FIM) [*Motor Subscale*]	93	Clinician Observation	E	General & Rehab
	BDPS [*Independence*]	52	Performance-based + Clinician Observation +/− Parent/other Interview	D	General
	Activities of Daily Living (ADL) ^4^	39	Unknown	Unknown	Unknown
	Modified Barthel Index (mBI)	30	Performance-based/Self/Parent/other Observation/Interview/Questionnaire	E	Adult Rehab
	VABS-2 parent report questionnaire [*Daily Living Skills Domain*]	30	Parent/other Questionnaire	D (2nd P, E)	General
	UK Functional Independence Measure and Functional Assessment Measure (UK FIM + FAM) [*Total Motor Subscore*]	12	Clinician Observation	E	Rehab
**Behavior**					
Social-Emotional	PEDI/PEDI-CAT [*Social Function/Social/Cognitive*]	573	Clinician Observation +/− Parent/other Questionnaire	E, D	General
	VABS-2 parent report questionnaire [*Socialization Domain*]	30	Parent/other Questionnaire	D (2nd P, E)	General
	DDST-II [*Personal-Social*]	20	Performance-based +/− Parent/other Interview	D	General
	BDI [*Social-Emotional*]	13	Clinician Observation +/− Parent/other Observation/Interview	D (2nd P, E)	General
	Strengths and Difficulties Questionnaire (SDQ)	8	Parent/other Questionnaire +/− Interview	D, E	General
	KSPD [*Language-Social*]	6	Clinician Observation	D, E	General
Adaptive Behavior	PEDI-CAT [*Responsibility*]	86	Parent/other Questionnaire	E, D	General
	VABS-2 parent report questionnaire [*Maladaptive Behavior Domain*]	30	Parent/other Questionnaire	D (2nd P, E)	General
	BDI [*Adaptive*]	13	Clinician Observation +/− Parent/other Observation/Interview	D (2nd P, E)	General
Executive Function	Behavior Rating Inventory of Executive Function (BRIEF)	7	Parent/other Questionnaire	D (2nd E)	General
**Brain Structure & Function**					
Neuroimaging	Magnetic resonance imaging (MRI); MRI with Diffusion tensor imaging (DTI)	525	Clinician Observation	D, P	General
	Positron emission tomography and computed tomography scan (PET-CT)	293	Clinician Observation	D, P	General
	Single photon emission computed tomography scan (SPECT)	111	Clinician Observation	D, P	General
Seizures/Electrical brain activity	Electroencephalogram (EEG)	255	Clinician Observation	D, P	General
	Seizure burden/frequency	128	Clinician Observation	D, P	General
**Cognition & General Development**					
Cognition and General Development	BSID-II [*Mental Scale*]	184	Performance-based	D (2nd E)	General
	WeeFIM [*Cognition*]	130	Clinician Observation +/− Parent/other Interview	E	Pediatric Rehab
	FIM [*Cognition Subscale*]	93	Clinician Observation	E	General & Rehab
	LAP-D [*Cognitive Skills*]	42	Performance-based/Parent/other Observation	D	General
	BDI [*Cognitive*]	13	Clinician Observation +/− Parent/other Observation/Interview	D (2nd P, E)	General
	UK FIM + FAM [*Total Cognitive Subscore*]	12	Clinician Observation	E	Rehab
	Wechsler Intelligence Scale for Children 4th/5th Edition (WISC-IV/-V)	7	Performance-based	D (2nd P)	General
	KSPD [*Cognitive-Adaptive*]	6	Performance-based	D, E	General
	Bayley Scales of Infant and Toddler Development 3rd Edition (BSID-III) [*Cognitive Scale*]	1	Performance-based	D (2nd P, E)	General
	Intelligence Quotient (IQ) Score	1	Performance-based	D (2nd P)	General
	Wechsler Preschool & Primary Scale of Intelligence 4th Edition (WPPSI-IV)	1	Performance-based	D (2nd P)	General
**Quality of Life**					
Quality of Life	Cerebral Palsy Quality of Life Questionnaire for Children (CP QOL-Child), Primary Caregiver Questionnaire	147	Parent/other Questionnaire	D, E	CP
	Child Health Questionnaire Parent Form 50 (CHQ)	94	Parent/other Questionnaire	D, E	General
	Short Form 8 (SF-8) Health Survey Quality of Life Questionnaire	1	Self/Parent/other Questionnaire	D, E	General
**Language & Communication**					
Language	Gesell Developmental Schedules	60	Clinician Observation	D	General
	LAP-D [*Speech skills*]	42	Performance-based/Parent/other Observation	D	General
	DDST-II [*Language Skills*]	20	Performance-based +/− Parent/other Interview	D	General
Communication	BDPS [*Communication*]	52	Performance-based + Clinician Observation +/− Parent/other Interview	D	General
	VABS-2 parent report questionnaire [*Communication Domain*]	30	Parent/other Questionnaire	D (2nd P, E)	General
	BDI [*Communication*]	13	Performance-based +/− Parent/other Observation/Interview	D (2nd P, E)	General
	Communication Function Classification System (CFCS)	11	Clinician Observation	C	CP
**Other**					
Other	Comprehensive Functional Assessment (CFA) Scale	94	Unknown	Unknown	Unknown
	Beery-Buktenica Developmental Test of Visual-Motor Integration 6th Edition	88	Performance-based	D	General
	Modified Rankin Scale (mRS)	30	Self (Clinician-led) Interview	E	Adult Rehab
	Caregiver Questionnaire Scale	14	Parent/other Questionnaire	Unknown	General
	CDCC Infant Mental Development Scale for general development status	1	Performance-based	D (2nd E)	General
	Mental Status Examination	1	Clinician Observation	D, E	General
**Safety**					
Safety	Safety reports/AEs/Routine laboratory and clinical assessments (including neuroimaging for safety exclusively)	1705	N/A	D	General
**Biomarkers**					
Biomarkers	Biomarkers (various)	144	N/A	D	General

Abbreviations: C, classification; CP, cerebral palsy; D, discriminative; E, evaluative; n, number of participants; P, predictive; Rehab, rehabilitation. ^1^ Does not include descriptive outcomes or investigator-developed, non-validated tools. ^2^ Number of participants across all studies assessed using instrument. ^3^ General motor could not be designated as either gross or fine. Includes oromotor function. ^4^ Review authors were unable to find information about this assessment.

## Data Availability

The data presented in this study are openly available in Open Science Framework https://osf.io/t9c8j/?view_only=9b82c37725834a1da1a50bb199cf5091 (accessed on 14 November 2022).

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
