# Peer review of "Are We Getting It Right? A Scoping Review of Outcomes Reported in Cell Therapy Clinical Studies for Cerebral Palsy"

_jcm, 2022, doi:10.3390/jcm11247319_

Round 1

Reviewer 1 Report

Thank you for the opportunity to evaluate this paper, which is a scoping review of reported outcomes from cell therapies for cerebral palsy.  In our view, this is a very important contribution to the field. 

 As cell therapies generate more and more enthusiasm, there is a need to ensure that the correct outcomes are measured.  This review shows that previous investigators focus a great deal on motor outcomes, while other important outcomes are reported less often and in less detail.  We found Figure 2 to be immensely helpful in illustrating this. 

The search strategy was appropriate and likely captured all relevant articles.  The data from those articles were appropriate.  Although it was not the focus of the paper, we thought it might be helpful to clarify in Table 1 if the reported ages were the reported ages at treatment of the study subjects, and how long the follow-up periods were.  In addition, we would be interested to know what kinds of biomarkers and safety data were reported, even just some examples of the more common ones.

Investigators engaged in cell-based research would do well to read this paper thoroughly as they design their trials.  In particular, we completely agree with the authors' conclusion beginning line 478:  "This highlights the importance of consumer engagement and co-design in medical research, to ensure that research, in particular clinical trials, are informed by community priorities, whilst remaining balanced with what scientists believe, and evidence tells us, cell therapies can feasibly achieve." 

Author Response

Response to Reviewer 1 Comments:

Point 1: Although it was not the focus of the paper, we thought it might be helpful to clarify in Table 1 if the reported ages were the reported ages at treatment of the study subjects, and how long the follow-up periods were.

Response to Point 1:

Added “at baseline” to header of column three in Table 1 to specify that ages specified were at baseline (typically just prior to cell intervention).

Added column to Table 1 to include the latest follow up timepoint for each included study. Note that many studies captured outcomes at multiple timepoints, and not all outcomes reported in Table 1 are assessed at the timepoint indicated. This information has been incorporated in a footnote in Table 1.

Point 2: In addition, we would be interested to know what kinds of biomarkers and safety data were reported, even just some examples of the more common ones.

Investigators engaged in cell-based research would do well to read this paper thoroughly as they design their trials.  In particular, we completely agree with the authors' conclusion beginning line 478:  "This highlights the importance of consumer engagement and co-design in medical research, to ensure that research, in particular clinical trials, are informed by community priorities, whilst remaining balanced with what scientists believe, and evidence tells us, cell therapies can feasibly achieve." 

Response to Point 2:

Added in details about Safety and Biomarkers:

“Reported safety data included adverse event reporting, routine laboratory and clinical assessments (e.g., bloods/biochemistry, x-ray), and neuroimaging conducted exclusively for safety.” LINES 361-363

“Of the four studies that conducted biomarker analysis these comprised assessment of various cytokine and growth factor levels including interferon (IFN)-γ, interleukin (IL)-17, IL-4, brain-derived neurotrophic factor (BDNF), and vascular endothelial growth factor (VEGF) [22], BDNF [23], pentraxin 3 (PTX-3), IL-8, and IL-10 [33] and PTX-3, IL-8, tumor necrosis factor (TNF)-α, and IL-1β [41].” LINES 367-372

Reviewer 2 Report

In my view a novel study, addressing an important question in a rapidly developing field. Study design and methods are well described with a pre published protocol. Writing is clear, Figures and Tables are well designed and helpful.

A minor issue is the possible need for a brief discussion on mechanisms of cell therapies, perhaps by a flow diagram. If a cell therapy achieves the holy grail of enhancing/improving/preserving neuronal function, then an improvment in motor function might plausibly be associated with a reduction in co-morbidities. If GMFM improves, co-morbidities should be reduced, not in a linear or predictable fashion but a link is to be expected. Therefore some of the differences in researcher vs parent/family goals may be more apparent than real and solved by education and better communication.

A few minor points, questions and suggestions:

1. Line 141: past tense might be better, were for are.

2. Were there differences between the outcome domains identified by the primary researchers vs the classification of outcome by the authors? Apart from the misuse of classifications (GMFCS, MACS and CFCS) as outcome measures, which the authors have dealt with admirably.

3. Figure 1: good PRISMA diagram. Here and elsewhere, I prefer the term "lost to follow up" rather than "dropout".

4. The term "single cell" is ambiguous and this sentence could be improved. I undestand the author's meaning but none of the studies used a "single cell"!

5. Table 1: excellent summary of complex information.

6. Line 253/4: This might be better expressed as "The source of cells was autologous....

7. Line 295: the term "bone fide" is not defined and is therefore somewhat ambiguous. Something like "psychometric qualities" might be better.

8. Fig 2 is excellent and communicates well.

9. Disappointing to see GMFCS MACS used as outcome measures but the authors have dealt with this very well in the discussion.

10. Line 382. "positive efficacy" perhaps "efficacy" is sufficient.

11. Paragraph 380-392: perhaps this is the point to link enhanced neuronal function to the expectation of reduced co-morbidities as an a priori assumption for researchers but not for parents/carers?

12. Line 401-402: The emphasis on safety is necessary because the therapies are novel. A brief description of Phase 1 vs Phase 2, 3 trials as per FDA might make this more clear for some readers. 

Author Response

Point 1: A minor issue is the possible need for a brief discussion on mechanisms of cell therapies, perhaps by a flow diagram. If a cell therapy achieves the holy grail of enhancing/improving/preserving neuronal function, then an improvment in motor function might plausibly be associated with a reduction in co-morbidities. If GMFM improves, co-morbidities should be reduced, not in a linear or predictable fashion but a link is to be expected. Therefore some of the differences in researcher vs parent/family goals may be more apparent than real and solved by education and better communication.

Response to point 1: Added text to introduction, discussion and a new Figure (Figure 3) to address this point.

 “The principal goal of cell therapies for the treatment of CP is remediation of the underlying brain injury thereby improving neuronal signaling, which could be achieved by either direct or indirect actions.” LINES 40-43

 “4.3. Mechanisms of Cell Therapies and Ensuing Effects

CP is caused by an interference, lesion, or abnormality of the developing brain which manifests as a disorder of movement and/or posture. Repairing the underlying brain injury, via direct or indirect mechanisms, to promote increased neuronal signaling and function is the aim of cell therapies for CP. As such, it is recognized that improvements in brain structure or connectivity following cell intervention could directly improve motor function. It is important to acknowledge however that links exist between motor skills and some comorbidities of CP. Figure 3 shows a schema of the proposed effects of stem cells for CP including therapeutic targets leading to remediation of the underlying brain injury, and resultant effects on various comorbidities, leading to the ultimate goal of improving quality of life. We wish to specifically highlight that changes in brain structure and connectivity producing improvements in motor function may have secondary effects on a number of motor-associated CP comorbidities (e.g., pain, sleep, drooling and speech). This may therefore mean that, in fact, improvements in various outcomes of importance to individuals with CP and their families may be more achievable than widely believed. In addition, the non-motor-associated comorbidities of CP (e.g., cognition, behavior) may be indirectly targeted by cell treatments.” LINES 628-646

PLUS Figure 3. Schematic representation of cell intervention effects and interlinked outcomes for CP including quality of life.

Point 2: Line 141: past tense might be better, were for are.

Response to point 2: “are” changed to “were”.

Point 3: Were there differences between the outcome domains identified by the primary researchers vs the classification of outcome by the authors? Apart from the misuse of classifications (GMFCS, MACS and CFCS) as outcome measures, which the authors have dealt with admirably.

Response to point 3: Not all primary researchers report (or perhaps consider) their measures in terms of outcome domains, for example primary researchers may simply refer to changes in scores on the Bayley-II, rather than stating that this outcome domain is overall motor. We devised the list of outcome domains based reviewing how each measure was described by test publishers and the associated domain scores obtainable. However, as mentioned in our limitations, some measures could conceivably be categorised differently. This would not necessarily be wrong as some measures do cover multiple outcome domains. We had to make these decisions to synthesise information and have transparently included our process in our methodology. Whilst this is an interesting question, we feel it is beyond the scope of this review to include in the discussion beyond how we have addressed this as a limitation.

Point 4: Figure 1: good PRISMA diagram. Here and elsewhere, I prefer the term "lost to follow up" rather than "dropout".

Response to point 4: Term “dropouts” has been changed to lost to follow up (LTFU). I have retained the detail regarding participants who withdrew before treatment.

Point 5: The term "single cell" is ambiguous and this sentence could be improved. I undestand the author's meaning but none of the studies used a "single cell"!

Response to point 5: “single cell intervention” changed to “one cellular intervention”

Point 6: Table 1: excellent summary of complex information.

Response point 6: Thank you

Point 7: Line 253/4: This might be better expressed as "The source of cells was autologous....

Response to point 7: Sentence was changed to “The source of cells was autologous in 32 studies (55%) and allogeneic in 25 studies (43%).”

Point 8: Line 295: the term "bone fide" is not defined and is therefore somewhat ambiguous. Something like "psychometric qualities" might be better.

Response to point 8: Changed “bona fide” to “true”

Point 9: Fig 2 is excellent and communicates well.

Response to point 9: Thank you

Point 10: Disappointing to see GMFCS MACS used as outcome measures but the authors have dealt with this very well in the discussion.

Response to point 10: We agree! Thank you

Point 11: Line 382. "positive efficacy" perhaps "efficacy" is sufficient

Response to point 11: Removed the word “positive”

Point 12: Paragraph 380-392: perhaps this is the point to link enhanced neuronal function to the expectation of reduced co-morbidities as an a priori assumption for researchers but not for parents/carers?

Response to point 12: New section added into Discussion (section 4.3) as detailed above.

Point 13: Line 401-402: The emphasis on safety is necessary because the therapies are novel. A brief description of Phase 1 vs Phase 2, 3 trials as per FDA might make this more clear for some readers

Response to point 13: Added text to provide additional details on clinical trial Phases and how these relate to capturing safety information. “Specifically, Phase 1, 2 clinical trials are important for understanding how a drug interacts with the human body, and to identify adverse events. Sub-sequent Phase 3 clinical trials, including larger numbers of participants, are important to show long-term or rare side effects.” LINES 495-498